# Validating a Portable Device for Blinking Analyses through Laboratory Neurophysiological Techniques

**DOI:** 10.3390/brainsci12091228

**Published:** 2022-09-10

**Authors:** Giulia Paparella, Alessandro De Biase, Antonio Cannavacciuolo, Donato Colella, Massimiliano Passaretti, Luca Angelini, Andrea Guerra, Alfredo Berardelli, Matteo Bologna

**Affiliations:** 1IRCCS Neuromed, Pozzilli, IS, 86077 Pozzilli, Italy; 2Department of Human Neurosciences, Sapienza University of Rome, 00185 Rome, Italy

**Keywords:** blinking movements, neurophysiology, portable device, kinematic analysis

## Abstract

Blinking analysis contributes to the understanding of physiological mechanisms in healthy subjects as well as the pathophysiological mechanisms of neurological diseases. To date, blinking is assessed by various neurophysiological techniques, including electromyographic (EMG) recordings and optoelectronic motion analysis. We recorded eye-blink kinematics with a new portable device, the EyeStat (Generation 3, blinktbi, Inc., Charleston, SC, USA), and compared the measurements with data obtained using traditional laboratory-based techniques. Sixteen healthy adults underwent voluntary, spontaneous, and reflex blinking recordings using the EyeStat device and the SMART motion analysis system (BTS, Milan, Italy). During the blinking recordings, the EMG activity was recorded from the orbicularis oculi muscles using surface electrodes. The blinking data were analyzed through dedicated software and evaluated with repeated-measure analyses of variance. The Pearson’s product-moment correlation coefficient served to assess possible associations between the EyeStat device, the SMART motion system, and the EMG data. We found that the EMG data collected during the EyeStat and SMART system recordings did not differ. The blinking data recorded with the EyeStat showed a linear relationship with the results obtained with the SMART system (r ranging from 0.85 to 0.57; *p* ranging from <0.001 to 0.02). These results demonstrate a high accuracy and reliability of a blinking analysis through this portable device, compared with standard techniques. EyeStat may make it easier to record blinking in research activities and in daily clinical practice, thus allowing large-scale studies in healthy subjects and patients with neurological diseases in an outpatient clinic setting.

## 1. Introduction

Eye blinking is characterized by the sequential opening and closing eyelid movements performed with the orbicularis oculi (OO) and levator palpebrae superioris (LPS) muscle [1,2,3]. There are three different types of blinking movements: (i) voluntary blinking, which occurs intentionally; (ii) spontaneous blinking, which occurs unconsciously at variable rates, and (iii) reflex blinking, which can be triggered by electrical supraorbital or corneal stimulation or auditory or visual stimuli [1,4,5,6,7]. The neural control of the OO and LPS muscles for the different types of blinking depends on several neuroanatomical substrates and partially overlapping circuitries in the lowest brainstem and cortical areas [1,3]. The anatomical framework of neural control of voluntary blinking is mainly centered on the frontal mesial areas [8,9]. Conversely, the spontaneous blink rate is related to dopamine levels in the central nervous system, particularly in the brainstem [1,6,10,11,12,13,14,15,16,17,18,19]. The reflex blinking neural control is primarily mediated by motor effectors and circuits at the brainstem level, though other supra-segmental structures, including the cerebral hemispheres, basal ganglia, and cerebellum, also modulate blink reflex circuit excitability [5,16,20,21,22]. The investigation of blinking provides a significant contribution to the physiological studies in healthy subjects. Furthermore, it has a role for the study of the pathophysiology of neurological diseases affecting the blinking circuits and neurodegenerative conditions, where disease, which is primarily localized to the basal ganglia, can secondarily lead to altered blinking movements through descending projections [1,6,10,11,12,13,14,15,16,23]. For example, a reflex blink analysis is very useful in brainstem disorders [20,24,25].

From an experimental standpoint, blinking is commonly assessed with various neurophysiological techniques. Electromyographic (EMG) techniques can be used to record blinking through surface electrodes [7,16]. Optoelectronic motion analyzer systems can also be used to perform a detailed objective evaluation of eyelid kinematics during blinking [6,10,11,12,13,14,15]. However, both EMG and optoelectronic motion analysis require expensive laboratory equipment, dedicated software, and trained personnel, precluding the large-scale applicability of these assessments. More recently, computer-assisted video acquisition systems or videography through smartphone cameras, suitable for the real-time measurement of blinking, even in the clinical setting, have been adopted [26,27,28,29,30,31]. In addition, recent technological advances have introduced the possibility of quantifying blinking using EyeStat (Generation 3, blinktbi, Inc., Charleston, SC, USA) (Figure 1), which provides blinking measurements with a portable device. However, no prior study has assessed the reliability and accuracy of the EyeStat device, compared with standard neurophysiological techniques.

In the present study, we recorded the eye-blink kinematics with EyeStat and compared these measurements with the kinematic data obtained with optoelectronic motion analysis and with the EMG measures collected through surface electrodes, a differential amplifier, and a digital-to-analog converter [6,10,11,12,13,14,15]. Demonstrating the accuracy and reliability of the blinking analysis through EyeStat would allow an easier way to record blinking in research activities and in daily clinical practice and would encourage the use of the device in large-scale studies on healthy humans and patients with neurological diseases in an outpatient clinic setting.

## 2. Materials and Methods

### 2.1. Participants

Sixteen healthy adult volunteers (13 males) aged 23 to 42 years (mean age ± 1 standard deviation—SD: 29.5 ± 4.75 years) participated in this study. None of them had neurological, psychiatric, eyelid, or eye disorders (including eyelid malposition or ocular surface pathology, dry eye syndrome, contact lens use, prior eyelid or intraocular surgery, prior periocular trauma, or craniofacial abnormalities). None of the participants was taking any medication that could potentially affect the central nervous system. All participants gave their informed consent and this study protocol was approved by the local ethics committee (DNU0221) and conducted in accordance with the Declaration of Helsinki. 

### 2.2. Experimental Design

We evaluated participants in one experimental session lasting about two hours. Voluntary, spontaneous, and reflex blinking movements were randomly recorded while participants were comfortably seated on a chair [6,10,11,12,13,14,15]. Since it was impossible to record blinking movements simultaneously with the EyeStat and SMART motion analysis system (BTS, Milan, Italy), due to the presence of the portable viewer, which would have interfered with the kinematic recordings, the three types of blinking were recorded in a sequential randomized order using the EyeStat device or SMART motion analysis system (Figure 1). However, to make sure that the experimental conditions were similar during the EyeStat and SMART recordings and that the two techniques were thus comparable, each blinking trial was also recorded by EMG.

Voluntary blinking was recorded by asking participants to look at a fixed point and to blink as fast as possible after a verbal command [6,10,11,12,13,14,15]. Following a brief practice session, two trials of eight voluntary blinks each were recorded with an interval of 4–5 s between each voluntary blinking movement. Spontaneous blinking was recorded for two 60 s epochs during a rest condition while participants were asked to relax and look straight at a fixed point [6,10,11,12,13,14,15]. The reflex blinking was evoked by delivering an air puff to the right or left cornea and to the eyelashes randomly and at random intervals during a recording trial of a 60 s duration while participants were looking at the fixed point [7]. In this case, the mechanical stimulation of the corneal surface activated Aδ and C free nerve endings in the cornea, eliciting a blink (‘corneal reflex’) [5,21]. Sixteen stimuli, eight for each eye, were delivered. During the reflex blinking recordings using the SMART motion system, the air puff stimuli were generated by a pressure unit, which was connected by a rigid plastic tube to a small buffer reservoir.

### 2.3. EyeStat Recordings and Analysis 

The EyeStat device (Figure 1) is a portable viewer in which the computer records high-speed video (280 frames/s) of the subject’s eyes with infrared cameras [32]. The air puffs initiated from a food grade carbon dioxide (CO_2_) cartridge delivered filtered air at 40 psi at prescribed and random intervals during the testing period and randomly to the right or left eye [33]. The device computer utilizes an eyelid tracking algorithm to track the subject’s right and left eyelids. The algorithm differentiated the voluntary, spontaneous, and reflex blinks, and the data from the eyelids was output along with the calculations of each blink’s characteristics. The typical testing sessions, from start to finish, occurred over a time frame of approximately 2 min. The blinking data analysis was performed using an automated algorithm, which provided information for the three types of blinking, including displacements (px) and upper eyelid velocity (px/sec), which were considered in the analysis.

### 2.4. Kinematic Recordings and Analysis

The SMART motion analysis system (Figure 1) is composed of three infrared cameras (120 Hz sampling rate) capable of following the displacement in the three-dimensional (3D) space of reflective markers of a negligible weight taped to the subject’s body segment of interest. Two markers were taped on the right upper eyelid and left upper eyelid [6,10,11,12,13,14,15]. Three additional markers were placed on the forehead and temporal level bilaterally, which enabled us to create a reference plane at the head level in order to exclude any contamination due to head motion during the eyelid movement. The analysis of the blinking data was performed using an automated algorithm, which provided a peak velocity and amplitude of the closing and opening phases for the three types of blinking movements [6,10,11,12,13,14,15]. Finally, we also considered the blink rate for the spontaneous blinking, i.e., the number of blinks per minute. 

### 2.5. EMG Recordings and Analysis

We recorded the EMG activity from the OO muscles of both sides using a pair of surface electrodes per side, with the active electrode on the lower eyelid and the reference electrode 3 cm away on the lateral canthus (Figure 1) [34,35]. A square electrode (32 × 32 mm) was positioned on the forehead as a ground electrode. The EMG raw signals were amplified and bandpass-filtered (20 Hz–3 kHz) by a Digitimer D360 amplifier (Digitimer Ltd., Welwyn Garden City, Herts, UK), that digitized at a sampling rate of 5 kHz (CED 1401 laboratory interface; Cambridge Electronic Design, UK), and was stored on a laboratory computer for online visual display. The data were analyzed offline with dedicated software (Signal software; Cambridge Electronic Design) [16,36]. 

### 2.6. Statistical Analysis

The OO-EMG tracks of the voluntary and spontaneous blinking collected during the blinking recordings with the EyeStat device and SMART system were compared by two different repeated measure analyses of variance (rmANOVA), using the factors TECHNIQUE (two levels: EyeStat device and SMART system) and RECORDING SIDE (two levels: right and left). When analyzing the OO-EMG, we added the factor STIMULATION SIDE (two levels: right and left). The voluntary and spontaneous blinking amplitudes from the right and left eye and the spontaneous blinking rate recorded with both the EyeStat device and SMART system were compared by two paired *t*-tests. We used the rmANOVAs to compare the reflex blinking amplitude from the right and left sides with the factors STIMULATION SIDE and RECORDING SIDE. Separate rmANOVAs for the EyeStat and SMART system recordings were used to compare the absolute velocity values of the opening and closing phases in the right and left eyes during voluntary and spontaneous blinking, with the factors BLINKING PHASE (two levels: opening and closing) and RECORDING SIDE. When evaluating the reflex blinking parameter, we included the factor STIMULATION SIDE and modified the analysis level for the factor RECORDING SIDE (two levels: ipsilateral and contralateral to the stimulated side). Pearson’s correlation was used to assess the possible relationships between the voluntary, spontaneous, and reflex blinking data obtained using the EMG, the EyeStat device and SMART system. Since no differences between sides emerged from the preliminary analysis of all techniques, the average of the right and left side eye was considered for the correlation analysis of voluntary and spontaneous blinking. For the reflex blinking correlation analysis, we averaged the recordings ipsilateral to the stimulated side. We then performed a post hoc power analysis for the correlation findings using the G*Power software [37]. *p* values <0.05 were considered statistically significant in all tests. The Tukey HSD test was used for post hoc analysis in the rmANOVAs. The Kolmogorov–Smirnov test was used to check if the data acquired using the EyeStat device and SMART system were normally distributed. Unless otherwise specified, all of the results are indicated as mean values ± 1 standard deviation (SD). The data were analyzed using STATISTICA^®^ (TIBCO Software Inc., Palo Alto, CA, USA).

## 3. Results

None of the participants experienced any adverse effects during or after the experimental session. All of the data recorded using the EyeStat device and SMART system were normally distributed (*p* always >0.05).

### 3.1. OO-EMG Analysis 

The voluntary and spontaneous blinking OO-EMG data collected during the EyeStat and SMART system recordings did not differ (Table 1). The analysis of the reflex blinking OO-EMG data showed, as expected, a significant RECORDING SIDE × STIMULATION SIDE (F_1, 15_ = 8.46, *p* = 0.01) interaction, indicating higher OO-EMG values recorded on the side ipsilateral to the stimulation (Table 1 and Table 2). However, no other significant factors or interactions emerged from the analysis (Table 2). Overall, these results indicate that the experimental conditions were similar during the EyeStat and SMART system recordings for the voluntary and spontaneous and reflex blinking movements, allowing us to compare the two techniques.

### 3.2. Voluntary Blinking

The analysis showed no difference in terms of eyelid displacement, e.g., amplitude, during voluntary blinking between the right and left side for the EyeStat and SMART system recordings (*p* = 0.30 and *p* = 0.38, respectively; Table 3). The rmANOVA on the velocity values showed a significant effect of the factor BLINKING PHASE, both for the EyeStat (F_1, 15_ = 66.31, *p* < 0.001) and SMART system recordings (F_1, 15_ = 70.51, *p* < 0.001), indicating higher velocity values for the closing phase, compared with the opening phase (Table 3 and Table 4). No other significant effects or interactions emerged from the data analysis of EyeStat or SMART system recordings (Table 4).

### 3.3. Spontaneous Blinking

There was no difference in the blinking rate between the EyeStat and SMART system recordings (EyeStat: 17.33 ± 8.54 [95% confidence interval-CI: 13.14–21.51]; SMART system: 16.06 ± 7.63 [95% CI: 12.32–19.8], *p* = 0.77). The eyelid displacement, e.g., the amplitude during the spontaneous blinking as recorded with the EyeStat device and SMART system, did not differ between the right and left sides (*p* = 0.55 and *p* = 0.84; Table 2). 

The rmANOVA on the velocity values during spontaneous blinking showed a significant effect of the factor BLINKING PHASE, both for the EyeStat (F_1, 15_ = 61.73, *p* < 0.001) and SMART system recordings (F_1, 15_ = 41.92, *p* < 0.001), with higher values observed for the closing phase, compared with the opening phase. However, neither analysis revealed any other significant factors or interactions for either the EyeStat or SMART system recordings (Table 4).

### 3.4. Spontaneous Blinking

There was no difference in the blinking rate between the EyeStat and SMART system recordings (EyeStat: 17.33 ± 8.54 [95% confidence interval-CI: 13.14–21.51]; SMART system: 16.06 ± 7.63 [95% CI: 12.32–19.8], *p* = 0.77). The eyelid displacement, e.g., the amplitude during the spontaneous blinking as recorded with the EyeStat device and SMART system, did not differ between the right and left sides (*p* = 0.55 and *p* = 0.84; Table 3). 

The rmANOVA on velocity values during spontaneous blinking showed a significant effect of the factor BLINKING PHASE, both for the EyeStat (F_1, 15_ = 61.73, *p* < 0.001) and SMART system recordings (F_1, 15_ = 41.92, *p* < 0.001), with higher values observed for the closing phase, compared with the opening phase. However, neither analysis revealed any other significant factors or interactions for either the EyeStat or SMART system recordings (Table 4).

### 3.5. Reflex Blinking

We found no significant factors or interactions in the analysis of the reflex blinking amplitude (Table 5). The rmANOVA on the velocity values of reflex blinking recorded with the EyeStat device showed a significant effect of the factor BLINKING PHASE (F_1, 15_ = 386.74, *p* < 0.001), with higher velocity values observed for the closing phase, compared with the opening phase (Table 3), as well as a significant RECORDING SIDE × STIMULATION SIDE interaction (F_1, 15_ = 21.21, *p* = 0.004), with higher velocity values being recorded in the side ipsilateral to the stimulation. No other significant factors or interactions emerged from the analysis (Table 5). 

The rmANOVA on the reflex blinking velocity values recorded with the SMART system showed, again, a significant interaction of the factor BLINKING PHASE (F_1, 15_ = 183.26, *p* < 0.001), with higher velocity values observed for the closing phase, compared with the opening phase, and significant RECORDING SIDE × STIMULATION SIDE (F_1, 15_ = 47.90, *p* < 0.001) and BLINKING PHASE × RECORDING SIDE × STIMULATION SIDE interactions (F_1, 15_ = 32.36, *p* < 0.001), overall indicating higher velocity values recorded in the side ipsilateral to the stimulation, which were more evident for the closing, compared with the opening phase. No other significant factors or interactions emerged from the analysis (Table 5). 

### 3.6. Correlation Analysis

The OO-EMG data collected during the EyeStat recordings correlated with the OO-EMG data collected during the SMART system recordings (r = 0.57, *p* = 0.02; r = 0.72, *p* = 0.002 and r = 0.83, *p* < 0.001 for voluntary, spontaneous, and reflex blinking respectively). This strengthened the concept that the experimental conditions were similar during the EyeStat and SMART system recordings and that the two methods were comparable. No other correlations were found between the OO-EMG data and EyeStat and SMART system parameters, including the blinking amplitude and velocity of the opening and closing phases (all *p* values > 0.05). For all of the blinking types, we found a significant linear correlation between the values of the eyelid displacement and velocity of the closing and opening blinking phases recorded with the EyeStat device and SMART system (r ranging from 0.85 to 0.57; *p* ranging from <0.001 to 0.02; power ranging from 0.67 to 0.99, Figure 2). Finally, the blinking rate measured with the EyeStat device significantly correlated with that measured with the SMART system (r = 0.83 *p* < 0.001) (Figure 3).

## 4. Discussion

In this study, we compared the blinking data recorded with the EyeStat device with data obtained with the SMART system and the surface EMG recordings. We found that the movement kinematics of voluntary, spontaneous, and reflex blinking recorded with the EyeStat device correlated with those obtained with the SMART system, i.e., with the kinematic recordings in 3D space, considered one of the current neurophysiological gold standards of blink assessment. Namely, for all three types of blinking, the amplitude and peak velocity of both the closing phase and the opening phase strongly correlated between these two techniques. We also confirmed the linear relationship in the spontaneous blink rate between the measurements obtained with EyeStat, the optoelectronic motion analysis, and the surface OO-EMG.

In interpreting the study results, we must consider some limitations and possible confounding factors. Firstly, we could not record simultaneously with the EyeStat device and SMART system. The EyeStat device consists of a visor that covers the eyelids of the participants examined, which makes it impossible to concomitantly analyze eyelid kinematics with the optoelectronic motion analysis system, which instead records movements in a 3D space through reflective markers placed on the participants’ eyelids. To overcome this possible limitation, we performed surface OO-EMG recordings during the EyeStat and SMART system recordings, which were then compared. The comparison showed no difference in the EMG activity during the EyeStat and SMART system recordings. Furthermore, we found a positive correlation between the OO-EMG data collected during the EyeStat recordings and the OO-EMG data collected during the SMART system recordings. Overall, these results allowed us to conclude that the level of OO muscle activation during the recordings performed with the two techniques was comparable, allowing us to then proceed with a correlation analysis of the collected data. Furthermore, there may be some blink measurement variability, possibly due to cognition, emotions, and psychological factors, including aging, fatigue, sleepiness, and the level of eye dryness [26,27,38,39,40,41,42]. However, we enrolled participants with a relatively narrow age range and we recorded the various blinking types in each individual during the same experimental session and at the same time of day, usually in the late morning or early afternoon. This methodological aspect minimized the effects of possible factors affecting the blinking variability. An even more important aspect to consider is that the recordings of the three recorded blink types, i.e., voluntary, spontaneous, and reflex, were randomized in each subject. We therefore consider it unlikely that our measurements were biased due to the recording sequence. Another limitation is the relatively limited number of participants studied. However, the use of quantitative analysis techniques allows accurate and reproducible data to be obtained. Furthermore, the participants were mostly males. Future studies on wider samples of healthy individuals, more homogenous in terms of gender distribution, compared with our sample, are needed to confirm the present results. Specifically concerning the spontaneous blinking rate, it has been repeatedly demonstrated that a 2 min recording provides reliable results [43]. Finally, we focused our analysis on the amplitude and velocity of movement. We did not analyze additional parameters in order to avoid redundant analyses considering that the relationship between amplitude, velocity, and duration of the down phase of a blink is well documented in humans [44].

Our study shows that the EyeStat device is accurate and precise in the study of blinking kinematics, as demonstrated by the strong correlation between EyeStat values and those obtained from other neurophysiological techniques. If confirmed by future studies on larger samples of healthy participants and patients with neurological and/or psychiatric conditions, our data may suggest that the EyeStat system may be suitable for numerous applications in neurology. First, it could be used for the evaluation of patients with possible brainstem damage [24]. Second, the EyeStat system could be effectively used, even by non-expert personnel, to record and analyze data obtained in patients with neurodegenerative diseases or psychiatric conditions, including parkinsonism and schizophrenia, where spontaneous blinking frequency is respectively reduced or increased [1,10,11,12,13,14,15,16,17,18,19,43,45,46]. Furthermore, due to the relationship between the central dopaminergic system and pain modulation [6,23,47,48], a further application of this system would be possible into conditions characterized by pain. Recent kinematic studies have also shown that voluntary blinking movements are altered in parkinsonism, specifically in patients suffering from atypical parkinsonism, compared with those with Parkinson’s disease [1,11,13,15]. Therefore, the routine use of blinking analysis in daily clinical practice, particularly voluntary blinking analysis, could allow the collection of valuable data for the differential diagnosis between these conditions. Finally, the EyeStat technology has previously been shown to demonstrate significant differences in several blink reflex parameters, including blink reflex latency, differential latency, the log of time to eyelid opening, and the log of number of oscillations between baseline scans and post-concussion scans among collegiate athletes [49].

Another interesting application of the study on blinking with portable devices would be the evaluation of the effects of treatments. The blinking parameters can be influenced by dopaminergic drugs and deep brain stimulation (DBS). For example, DBS has a detrimental effect on voluntary blinking kinematics, possibly inducing the so-called phenomenon of apraxia of the eyelid opening [12,50,51]. Therefore, the systematic study of voluntary blinking could provide useful data to monitor the medium- and long-term effects of drugs and surgery. Finally, the ability to analyze blinking movements in a 3D space in 2D by video recording analysis makes it possible to study the eyelid opening phase, which to date can be only studied with EMG by inserting an electrode needle into the LPS muscle. This is relevant, since the analysis of the palpebral opening phase represents an indirect and non-invasive way to study the neuroanatomical substrates of this movement which involve brainstem nuclei and areas thought to be related to the control of pain and autonomic functions, mainly the periaqueductal gray [6,13,52,53,54].

## 5. Conclusions

We have validated the blinking recordings with a new portable device for blink analysis, EyeStat, and compared the results obtained from this device with those obtained using the current gold standard. In the context of neurophysiological techniques commonly used to analyze and record blinking, the EyeStat device has attractive benefits, particularly in terms of its speed of use and data analysis, even for personnel with little experience in experimental research. These advantages make the EyeStat device suitable for large-scale studies on several neurological disorders. Also, taking into consideration that the EyeStat device may provide several parameters, including the latencies of reflex blinking, EyeStat may be useful in clinical practice for the evaluation of patients with various neurological conditions, including primary brainstem and neurodegenerative diseases, particularly in terms of diagnosis and disease monitoring over time.

## Figures and Tables

**Figure 1 brainsci-12-01228-f001:**
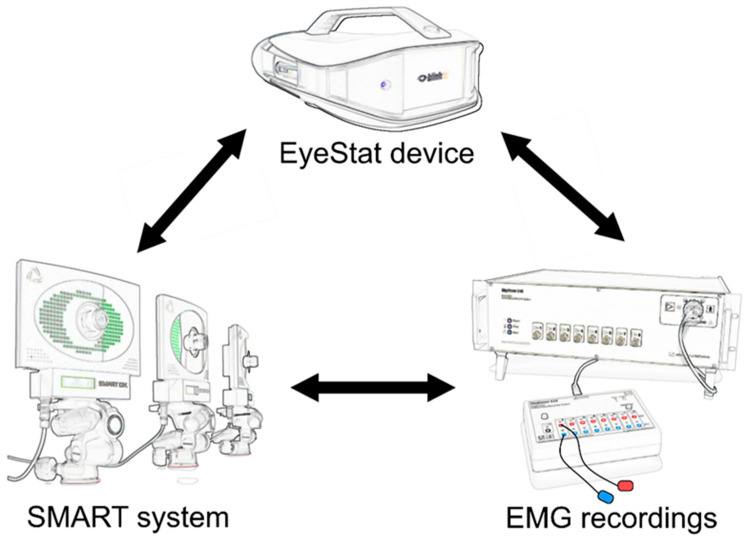
**Methods used to record blinking**. Three methods were used to record voluntary, spontaneous, and reflex blinking. The EyeStat device (Generation 3, Blinktbi, Charleston, SC, USA) and the SMART motion analysis system (BTS, Milan, Italy) were applied in a randomized order. An electromyographic (EMG) recording system was also used to simultaneously record every blinking movement trial.

**Figure 2 brainsci-12-01228-f002:**
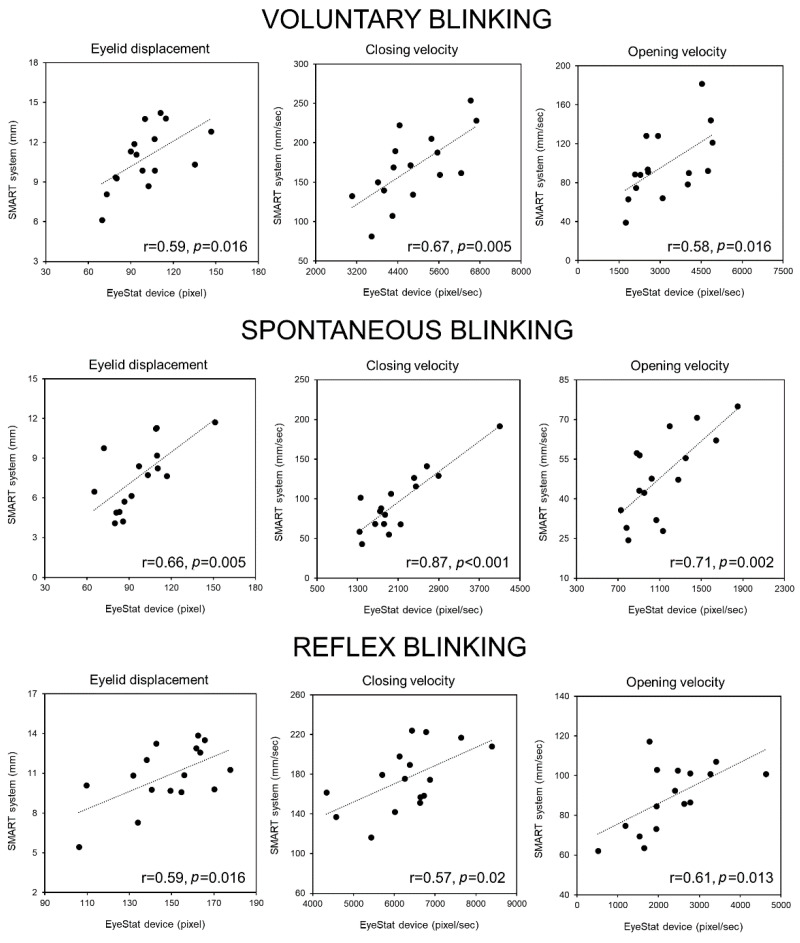
**Correlations between the various blinking parameters measured with the two different methodologies.** The amount of eyelid displacement and the velocity of the closing and opening phases of blinking measured with the EyeStat device during voluntary (top row), spontaneous (middle row), and reflex blinking (bottom row) significantly correlated with values of the same measures obtained using the SMART system (r and *p* values are depicted in the lower right corner).

**Figure 3 brainsci-12-01228-f003:**
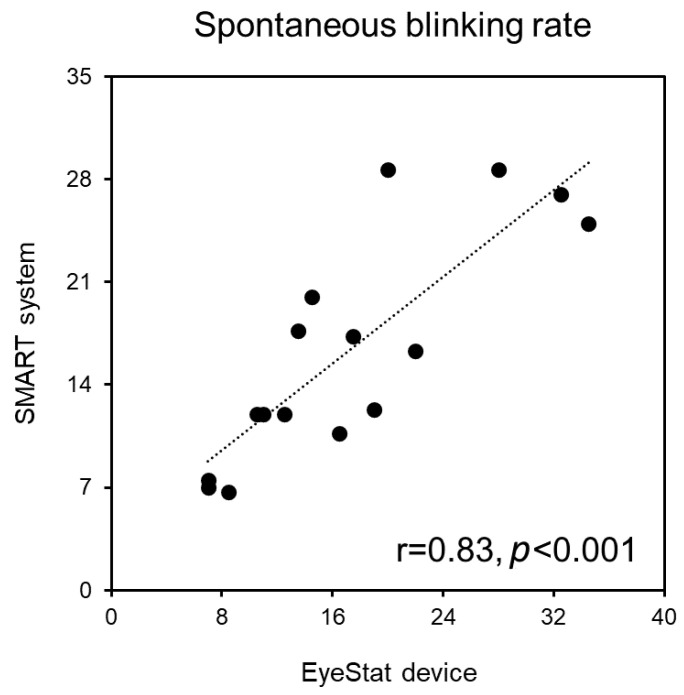
**Correlation between the blink rate as measured with the two different methodologies.** The axes display the blink rate, as measured by the number of spontaneous blinks per minute. The blink rate recorded during spontaneous blinking with the EyeStat device significantly correlated with that recorded using the SMART system.

**Table 1 brainsci-12-01228-t001:** **Orbicularis oculi muscle electromyographic recordings.** Electromyographic (EMG) recordings are expressed in mV*ms. Data are indicated by mean ± standard deviation of the mean.

	Voluntary		Spontaneous		Reflex			
	Right	Left	Right	Left	Stimulation Right		Stimulation Left	
					Right	Left	Right	Left
**EyeStat**	6.42 ± 6.64	6.64 ± 6.7	2.14 ± 0.96	2.18 ± 1.11	7.77 ± 4.85	7.63 ± 6.39	6.38 ± 4.24	7.89 ± 5.63
**SMART**	7.18 ± 4.52	6.47 ± 3.36	2.3 ± 1.17	2.13 ± 1.03	6.58 ± 7.04	4.37 ± 3.93	4.26 ± 4.04	7.34 ± 8.06

**Table 2 brainsci-12-01228-t002:** **Statistical analysis for the orbicularis oculi muscle electromyographic recordings.** The table shows the statistical output from the rmANOVA performed on the electromyographic data during voluntary, spontaneous and reflex blinking. REC SIDE: recording side; STIM SIDE: stimulation side. Significant factors and interactions are depicted in bold.

Statistical Factors	Voluntary	Spontaneous	Reflex
TECHNIQUE	F_1, 15_ = 1.39, *p* = 0.25	F_1, 15_ = 0.08, *p* = 0.77	F_1, 15_ = 1.72, *p* = 0.20
REC SIDE	F_1, 15_ = 1.13, *p* = 0.30	F_1, 15_ = 0.31, *p* = 0.58	F_1, 15_ = 0.50, *p* = 0.48
STIM SIDE	-	-	F_1, 15_ = 0.49, *p* = 0.82
TECHNIQUE × REC SIDE	F_1, 15_ = 2.49, *p* = 0.13	F_1, 15_ = 3.44, *p* = 0.08	F_1, 15_ = 0.009, *p* = 0.92
TECHNIQUE × STIM SIDE	-	-	F_1, 15_ = 0.59, *p* = 0.45
REC SIDE × STIM SIDE	-	-	**F_1, 15_ = 8.46, *p* = 0.01**
TECHNIQUE × REC SIDE × STIM SIDE	-	-	F_1, 15_ = 2.77, *p* = 0.11

**Table 3 brainsci-12-01228-t003:** **Kinetic parameters of the voluntary, spontaneous, and reflex blinking measures with the EyeStat and SMART systems.** Open Vel: velocity of the opening phase; Clos Vel: velocity of the closing phase. Amplitude is expressed in pixels (px) when measured with EyeStat and in millimeters (mm) when measured with the SMART system. Absolute blinking velocity values are expressed in px/s when measured with EyeStat and in mm/s when measured with SMART. Data are indicated by mean ± standard error of the mean. Confidence intervals of 95% for the amplitude means are reported indicated in square brackets.

	Voluntary		Spontaneous		Reflex			
	Right	Left	Right	Left	Stimulation Right		Stimulation Left	
					Right	Left	Right	Left
**EyeStat**								
Amplitude	104.59 ± 28.77 [90.49–118.69]	95.21 ± 25.84 [82.46–107.78]	99.38 ± 24.12 [87.56–111.2]	94.61 ± 20.69 [84.47–104.75]	146.12 ± 23.25	143.39 ± 23.26	144.87 ± 19.32	147.44 ± 19.87
Open Vel	3351.1 ± 1365.02	2987.09 ± 1094.06	1169.7 ± 346.12	1069.64 ± 316.65	2513.31 ± 1237.23	2149.15 ± 651.69	2356.94 ± 885.02	2164.35 ± 824.53
Clos Vel	5061.37 ± 1254.7	4614.09 ± 1300	2105.47 ± 719.53	2068.22 ± 709.8	6436.65 ± 1153.66	5819.11 ± 1191.43	6107.45 ± 1143.32	6111.58 ± 969.33
**SMART**								
Amplitude	11.1 ± 2.75 [9.75–12.45]	10.71 ± 2.85 [9.31– 12.11]	7.64 ± 2.78 [6.28–9]	5.13 ± 2.54 [3.88–6.37]	10.47 ± 2.69	9.76 ± 2.2	10.86 ± 2.3	10.84 ± 2.38
Open Vel	100.5 ± 35.2	89.72 ± 36.29	48.29 ± 15.81	48.59 ± 18.78	91.63 ± 18.84	77.31 ± 16.96	82.13 ± 17.73	86.56 ± 20.13
Clos Vel	170.38 ± 46.57	166.44 ± 46.23	96.1 ± 39.27	94.96 ± 41.05	187.46 ± 41.65	155.81 ± 40.27	155.41 ± 33.74	176.7 ± 33.91

**Table 4 brainsci-12-01228-t004:** **Statistical analysis for the EyeStat device and SMART system recordings (voluntary and spontaneous blinking).** The table shows the statistical output from the rmANOVA performed on the EyeStat device and SMART system on the voluntary and spontaneous blinking data. PHASE: blinking phase; REC SIDE: recording side. Significant factors and interactions are depicted in bold.

Statistical Factors	Voluntary EyeStat (Velocity)	Voluntary SMART (Velocity)	Spontaneous EyeStat (Velocity)	Spontaneous SMART (Velocity)
PHASE	**F_1, 15_ = 66.31, *p* < 0.001**	F_**1, 15**_ **= 70.51, *p* < 0.001**	F_**1, 15**_ **= 61.73, *p* < 0.001**	F_**1, 15**_ **= 41.92, *p* < 0.001**
REC SIDE	F_1, 15_ = 2.49, *p* = 0.13	F_1, 15_ = 3.42, *p* = 0.07	F_1, 15_ = 2.47, *p* = 0.14	F_1, 15_ = 0.16, *p* = 0.90
PHASE × REC SIDE	F_1, 15_ = 0.08, *p* = 0.77	F_1, 15_ = 0.82, *p* = 0.38	F_1, 15_ = 2.79, *p* = 0.11	F_1, 15_ = 0.35, *p* = 0.85

**Table 5 brainsci-12-01228-t005:** **Statistical analysis for the EyeStat device and SMART system recordings (reflex blinking).** The table shows the statistical output from the rmANOVA performed on the EyeStat device and SMART system on the reflex blinking data. PHASE: blinking phase; REC SIDE: recording side; STIM SIDE: stimulation side. Significant factors and interactions are depicted in bold.

Statistical Factors	Reflex EyeStat (Velocity)	Reflex EyeStat (Amplitude)	Reflex SMART (Velocity)	Reflex SMART (Amplitude)
PHASE	**F_1, 15_ = 386.74, *p* < 0.001**	-	**F_1, 15_ = 183.26, *p* < 0.001**	-
REC SIDE	F_1, 15_ = 3.88, *p* = 0.07	F_1, 15_ = 0.03, *p* = 0.87	F_1, 15_ = 1.50, *p* = 0.24	F_1, 15_ = 1.29, *p* = 0.27
STIM SIDE	F_1, 15_ = 0.39, *p* = 0.53	F_1, 15_ = 0.47, *p* = 0.50	F_1, 15_ = 0.51, *p* = 0.49	F_1, 15_ = 3.33, *p* = 0.09
PHASE × REC SIDE	F_1, 15_ = 0.11, *p* = 0.74	-	F_1, 15_ = 0.01, *p* = 0.95	-
PHASE × STIM SIDE	F_1, 15_ = 0.43, *p* = 0.52	-	F_1, 15_ = 2.69, *p* = 0.12	-
REC SIDE × STIM SIDE	**F_1, 15_ = 21.21, *p* = 0.004**	F_1, 15_ = 2.43, *p* = 0.14	**F_1, 15_ = 47.90, *p* < 0.001**	F_1, 15_ = 1.37, *p* = 0.26
PHASE × REC SIDE × STIM SIDE	F_1, 15_ = 2.31, *p* = 0.14	-	**F_1, 15_ = 32.36, *p* < 0.001**	-

## Data Availability

Data that support the findings of this study are available upon request to the corresponding author.

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
