# Peer review of "Validating a Portable Device for Blinking Analyses through Laboratory Neurophysiological Techniques"

_brainsci, 2022, doi:10.3390/brainsci12091228_

Round 1

Reviewer 1 Report

I appreciated having the opportunity to review this manuscript. This study compared blinking data recorded with the EyeStat device with data obtained with the SMART system and surface EMG recordings, and found that movement kinematics of voluntary, spontaneous, and reflex blinking recorded with the EyeStat device correlated with those obtained with the SMART system. Also the amplitude and peak velocity are strongly correlated between these two techniques. This manuscript is well written and organized, and  I only have some comments and suggestions: 

1. Please specify the reason of only male volunteers were included, also only healthy volunteers were included in this study. Only male health volunteers should not be representative for the potential population this device will be applied to in the practice. Is there any plan in the future studies that include female volunteers and patients with neurological, psychiatric, and other disorder? 

2. Since there were only 13 subjects, I am wondering whether or not the rmANOVA assumptions have been met by the data.

3. Captions for tables should be placed above the table.

4. It is better to combine tables 1-2 together to convey clearer information. For example, mean+/SD for groups and their corresponding test statistic and p value.

5. Post hoc power analysis was conducted in the study, but the corresponding results are not present. In addition, please explain why used two adjustment methods ( bonferroni and tukey), but not used one for all the analyses. 

Author Response

Reviewer 1

I appreciated having the opportunity to review this manuscript. This study compared blinking data recorded with the EyeStat device with data obtained with the SMART system and surface EMG recordings, and found that movement kinematics of voluntary, spontaneous, and reflex blinking recorded with the EyeStat device correlated with those obtained with the SMART system. Also the amplitude and peak velocity are strongly correlated between these two techniques. This manuscript is well written and organized, and I only have some comments and suggestions:

  1. Please specify the reason of only male volunteers were included, also only healthy volunteers were included in this study. Only male health volunteers should not be representative for the potential population this device will be applied to in the practice. Is there any plan in the future studies that include female volunteers and patients with neurological, psychiatric, and other disorder?

We thank the reviewer for the positive comments.

In the present study, we actually enrolled 16 healthy adult volunteers, including 13 males and 3 females. We only studied healthy subjects since this was a pilot study aiming to investigate the accuracy and reliability of blinking analysis through EyeStat device. Future studies on wider samples of healthy individuals, more homogenous in terms of gender distribution as compared the sample we studied, are needed to confirm the present results, and to lay the groundwork for future large-scale research in which Eyestat will be used in in patients with neurological and psychiatric disorders. Following the reviewer suggestion, in the present version of the manuscript we have empathized these issues on page 11, lines 386-388 and lines 396-398.

  1. Since there were only 13 subjects, I am wondering whether or not the rmANOVA assumptions have been met by the data.

We acknowledge the reviewer comment. The rmANOVAs represent the most appropriate analyses to detect any overall differences in mean scores under three or more different conditions, and the assumptions have been met by the data.

  1. Captions for tables should be placed above the table.

We have now placed the captions above the tables.

  1. It is better to combine tables 1-2 together to convey clearer information. For example, mean+/SD for groups and their corresponding test statistic and p value.

We thank the reviewer for this comment. However, since the two table depict different data, we preferred to keep them apart.  

  1. Post hoc power analysis was conducted in the study, but the corresponding results are not present. In addition, please explain why used two adjustment methods (bonferroni and tukey), but not used one for all the analyses.

The results of the post hoc power analysis are detailed on page 8, line 294: the analysis showed that the power ranged from 0.67 to 0.99. Concerning the post hoc analysis correction, we have performed the analyses using one adjustment methods, i.e., the Tukey HSD test (Page 4, lines 179).  The study results did not change.

Reviewer 2 Report

Dear Authors, I wish to submit the review of the paper titled: "Validating a portable device for blinking analysis through laboratory neurophysiological techniques."

The research design is adequate, and the paper is well written and provides sufficient information according to the Current Literature. The Authors should be commended for their work.

Despite the excellent paper, some points require attention:

1. Could you please add the information regarding the sample size? Were the data normally distributed?

2. "Our data suggest that the EyeStat system may be suitable for numerous applications in neurology."  The Discussion and Conclusions do not sufficiently consider the certainty of the body evidence (results) in drawing conclusions. Testing affected patients may be helpful for the conclusions stated.

3. Could you please add information regarding the cost of this device and the impact on clinical practice? (Section Discussion)

Author Response

Dear Authors, I wish to submit the review of the paper titled: "Validating a portable device for blinking analysis through laboratory neurophysiological techniques."

The research design is adequate, and the paper is well written and provides sufficient information according to the Current Literature. The Authors should be commended for their work.

Despite the excellent paper, some points require attention: 

  1. Could you please add the information regarding the sample size? Were the data normally distributed?

We thank the reviewer for the comments. Since this was a pilot study, we did not perform any a priori sample size estimation. However, we provided a post hoc power analysis for the main study outcomes, i.e., the correlation analysis results, showing that the power ranged from 0.67 to 0.99.

All data recorded with the EyeStat device and the SMART system were normally distributed, as assessed by the Kolmogorov-Smirnov test (p always >0.05). We have now added this information in the manuscript (page 4, line 179-181 and page 5, lines 188-189).

  1. "Our data suggest that the EyeStat system may be suitable for numerous applications in neurology."  The Discussion and Conclusions do not sufficiently consider the certainty of the body evidence (results) in drawing conclusions. Testing affected patients may be helpful for the conclusions stated.

Following the reviewer suggestion, we have slightly modified the study discussion and softened the part on the possible applications of Eyestat in neurology, by underling that our results on healthy participants need to be confirmed on patients affected by neurological and/or psychiatric conditions (see page 11, lines 386-388, 396-398, and lines 399, 405, and 415).

  1. Could you please add information regarding the cost of this device and the impact on clinical practice? (Section Discussion)

The EyeStat Generation 3 device is not for commercial sale. It is intended for clinical research purposes only.

Round 2

Reviewer 2 Report

The authors' changes met the requirements and expectations.